# Low coverage of HIV testing among adolescents and young adults in Nigeria: Implication for achieving the UNAIDS first 95

Anthony Idowu Ajayi[1]*, Oluwafemi Emmanuel Awopegba[2], Oluwafemi Atanda Adeagbo[3,4], Boniface Ayanbekongshie Ushie[1]

1 Population Dynamics and Reproductive Health and Rights Unit, African Population and Health Research Center, Kitisuru, Nairobi, Kenya, 2 Economics and Business Policy Department, Junior Research Fellow, Nigerian Institute of Social and Economic Research, Ibadan, Nigeria, 3 Department of Sociology, University of Johannesburg, Johannesburg, South Africa, 4 Africa Health Research Institute, Mtubatuba, South Africa

* ajayianthony@gmail.com

**Data Availability Statement:** The MICS 5 data for Nigeria underlying the results presented in the study are freely available from UNICEF at http://mics.unicef.org/surveys.

## Abstract

### Background

Most studies on HIV testing among young people in Nigeria are not nationally representative. As such, recent nationally representative data, such as the Multiple Indicator Cluster Survey (MICS), could help assess the current level of HIV testing among young people, a key target population for HIV prevention in the country. In this study, we examined the coverage and factors associated with HIV testing among adolescents and young adults (AYA).

### Methods

We used the data for 14,312 AYA that examined recent and lifetime HIV testing from the 2017 MCIS. Our outcomes of interest were ever tested for HIV and recently tested for HIV. We examined the association between socio-demographic factors (e.g., age, marital status, education attainment, wealth status), stigma belief, exposure to media and HIV knowledge, and uptake of HIV testing using adjusted and unadjusted logistic regression models.

### Results

Less than a quarter of the AYA (23.7%) had ever tested for HIV, and an even lower proportion (12.4%) tested in the year preceding the survey. More females (25.4%) compared to males (20.8%) had ever tested for HIV. Young people who were aged 20–24 years (AOR 1.52, 95% CI 1.34–1.72), married (AOR 2.42, 95% CI 1.98–2.97), had higher educational attainment (AOR 5.85, 95% CI 4.39–7.81), and belonged to the wealthiest quintile (AOR 1.99, 95% CI 1.53–2.60), had higher odds of having ever tested for HIV compared to those aged 15–19 years, never married, had no formal education and belonged to the poorest wealth quintile. Also, those who had positive stigma belief towards people living with HIV (AOR 2.93, 95% CI 2.47–3.49), had higher HIV knowledge (AOR 1.62, 95% CI 1.24–2.11), and higher media exposure (AOR 1.64, 95% CI 1.36–1.97), had higher odds of having ever

**Funding:** The author(s) received no specific funding for this work.

**Competing interests:** The authors have declared that no competing interests exist.

tested compared to those who had more negative stigma belief, had low knowledge of HIV and low media exposure.

## Conclusion

The HIV testing coverage among AYA in Nigeria is well below the national target of 95% indicated in the national HIV/AIDS strategic framework (2017–2021). Also, the low rate of HIV testing found in this study means realising the UNAIDS first 95 will require interventions targeting AYA. These interventions should focus on improving young people's knowledge of HIV, reducing negative stigma belief through media campaigns and increasing access to HIV testing through home-based testing and "opt-out" strategy at the point of care.

## Background

Despite the significant impact of antiretroviral therapy (ART) and other preventive strategies in the reduction of new HIV infections and AIDS-related deaths, HIV remains the leading cause of deaths among adolescents and young adults in sub-Saharan Africa (SSA)[1, 2]. Close to 66% of the 5,000 new HIV infection a day in SSA. Late diagnosis and poor adherence to ART and retention in care are among the main reasons for poor treatment outcomes in this age cohort. There is evidence that less than a quarter of young adults in sub-Saharan Africa have ever tested for HIV, and the rate is even much lower among young men, and for those who are HIV+, the testing is usually performed late[3–6]. Early diagnosis of HIV is a critical first step to initiating ART, preventing new HIV infections and AIDS-related deaths, and ensuring good quality of life and wellbeing for young people potentially living with HIV[1].

Adolescents and young adults (AYA) (15–24 years) bear a disproportionate burden of HIV, particularly for new HIV infection[1]. Approximately 510,00000 [300,000–740,000] AYA were newly infected with HIV in 2018[7] and 61% of these new infection occurred in SSA[1]. What is more, adolescent girls in SSA account for four in five new infections[7]. Also, AYA are less likely to have tested for HIV, have the lowest level of adherence to ART and viral load suppression[1]. For example, a study shows that only 20.7% adolescents in South Africa achieved complete adherence compared with 40.5% among adults[8]. The study further reveals that 43.6% of adolescents achieved viral suppression at 24 months of follow up compared with 62.3% among adults[8]. Screening for HIV is an important public health intervention required to reduce the burden of HIV, especially among AYA who are most at risk of contracting the infection[9]. HIV testing is beneficial not only for individuals that test positive but also for those who return a negative result given that uptake of HIV testing promotes preventive behaviours [10, 11]. The World Health Organisation (WHO) recommended "Universal test and treat" as the approach for eliminating new HIV infections [12].

Recognising the centrality of HIV testing to averting new infections, the UNAIDS set the ambitious 95-95-95 target towards ending the HIV epidemic by 2030. The target aims to ensure that by 2030, 95% of all people living with HIV will know their HIV status, 95% of all people with diagnosed HIV infection will receive sustained antiretroviral therapy, and 95% of those on treatment will have suppressed viral load[13]. A newly published study in the New England Journal of Medicine[9] has established the effectiveness of the "universal test and treat" in reducing new HIV infection, further underscoring the need for scaling up of HIV testing. Nigeria needs to scale up the universal test and treat, especially among young people who are most at risk[9], to achieve a significant reduction in the rate of new infections.

Several studies have focused on factors hindering the uptake of HIV testing with lack of access, fear of being diagnosed positive, perceived low risk of contracting HIV, fear of stigmatisation, and the perceived psychological burden of living with HIV being among the most reported barriers [5, 14–19]. In contrast, physical health deterioration, knowing someone who died as a result of HIV, knowing one's partner HIV status, HIV knowledge, couples' open communication about HIV, expanding access to HIV testing and treatment, as well as having a social support network and the guarantee of anonymity and confidentiality are factors that facilitate HIV testing uptake [5, 20–22].

At 1.5%, Nigeria has a low prevalence of HIV, but the country's large population of >200 million[23] means that the number of people living with HIV in the country is substantial [24, 25]. A significant proportion (up to 34.1%) of new infections occurs among adolescents and young adults in the country[2]. Lack of comprehensive sexual health education, sexual and gender-based violence, poor access to sexual and reproductive health services, and poverty are among the drivers of new HIV infections in the country [26]. The lack of access to HIV testing, especially in rural areas, is also among the factors fueling new HIV infection [27]. However, in the past decade, significant effort has been made towards scaling up of access to HIV testing in the country [28, 29]. A study examining the trend in HIV testing among AYA from 2003 to 2013 shows that testing uptake has increased from 14.6% among males to about 22% and from 5.9% to 19.4% among females [4]. Nevertheless, the level of HIV testing uptake among young people in Nigeria is low compared to countries like Kenya, Zambia, and South Africa [4]. What is more, only 38% of people living with HIV (PLHIV) are aware of their status in Nigeria, which has harmful consequences for new infections, treatment outcomes, and cost of care.

Most studies on HIV testing among young people in Nigeria are not nationally representative or are now outdated [4, 5, 19, 30]. As such, recent nationally representative data, such as the Multiple Indicators Cluster Survey (MICS), could help assess the current level of HIV testing among young people and the level of progress recorded since the scaling up of HIV testing services in Nigeria. Given this context, the present study aims to determine the level of HIV testing among adolescents and young adults in Nigeria. Also, the study examines the influence of demographics (such as age, sex, education, knowledge of HIV), behavioural (sexual activity, and condom use), community/social (knowledge, stigma and media exposure) and contextual factors (household wealth, geopolitical zones and urban/rural residence) on uptake of HIV testing.

Health behavioural theories, like the Health Belief Model, have highlighted the role of knowledge as a critical factor influencing the use of services[31–34]. Thus, we hypothesise that knowledge of HIV is associated with increased odds of uptake of HIV testing. With improved knowledge of HIV, individuals can assess their risk of contracting HIV, access HIV testing, and initiate ART. Furthermore, better knowledge of HIV could help reduce stigma behaviour, which is a significant barrier to uptake of HIV testing in settings in SSA [14, 18, 35]. Advances in HIV treatment now mean HIV is no longer a death sentence and with undetectable viral load (undetectable means untransmittable), implying zero risks of HIV transmission [36]. Spreading this message is critical to reducing HIV stigma and increase uptake of HIV testing. Our primary proposition in this study is that expanding access to HIV testing alone, without improving HIV knowledge, and addressing internalised HIV-related stigma, is not enough to increase uptake of HIV testing. In other words, focusing on improving HIV knowledge of young people is as important as expanding access to HIV testing. This study aligns with the global focus on meeting the UNAIDS' first 95', which aims to ensure 95% of people living with HIV know their status by 2020 and which Nigeria aims to achieve by 2021. Achieving this aim in Nigeria will require evidence-informed policies.

## Materials and methods

The data used in this study were from the fifth multiple indicators cluster survey (MICS) for Nigeria, which is freely available upon request at http://mics.unicef.org/surveys. The 5th multiple cluster survey was carried out from September 2016 to January 2017 throughout the country. The MICS was a cross-sectional study conducted across all the geopolitical zones in 37, 440 households in Nigeria. The sample size was representative of the country and also of the six geopolitical zones.

### Sampling methods

The survey adopted a multi-stage cluster sampling approach, designed to give reliable estimates for numerous indicators on men, women and children's wellbeing, at national, urban/rural, geopolitical zone and state levels in the country. The states were identified as the sampling strata, while the Enumeration Areas (EAs) were the Primary Sampling Units (PSUs). At the first stage, 60 EAs were selected systematically within each (34) states and Abuja (FCT), while 120 EAs each were selected from Kano and Lagos states. In the second stage, a systematic sample of 16 households was taken from each of the selected EAs. Out of 37,440 households sampled, 36,176 women of reproductive age (15–49 years) and 17,868 men were interviewed.

### Ethical statement

We did not need to seek ethical approval for the study given that this is a secondary analysis of publicly available de-identified data.

### Study population

There were 52,690 MICS respondents (16,514 men and 36,176 women). However, the focus of this study was AYA 15–24 yrs, who comprised 18,494 (35.1% out of all respondents), who met the selection criteria. The analysis was further limited to 14, 312 AYA who responded to the questions on HIV testing. the overview of the study sample is presented in Fig 1.

### Variables and measurements

Our dependent variables of interest in this study are two-category nominal measure of uptake of HIV testing. We used two different measures of HIV testing, focusing on having ever tested for HIV and recency of HIV testing (defined as the uptake of HIV test in the past year). The responses were dichotomised into Yes or No with yes coded as "1" and no coded as "0". Participants who tested for HIV in the previous year before the survey were categorised as recently tested for HIV and assigned "1" and those who did not were assigned "0".

We have three sets of explanatory variables, individual-level factors (demographics, behavioural (sexual activity, and condom use, knowledge, stigma, and media exposure), household-level factor (household wealth) and contextual factors (geopolitical zones and urban/rural residence). The demographic variables are age (15–19 and 20–24), sex (male and female), marital status (currently married, formerly married and never married), and level of education (no formal education, primary, secondary and tertiary). The behavioural variables are "ever had sex" and "condom use", which have dichotomous responses coded as "1" if reported Yes, and "0" if No.

The knowledge of HIV was constructed to measure the respondents' awareness about the virus, transmission routes, and how to prevent HIV transmission. Seven relevant questions were contained in the MCIS and were used to assess participants' knowledge of HIV. Each respondent was given a score between 0 to 7, depending on the number of questions answered

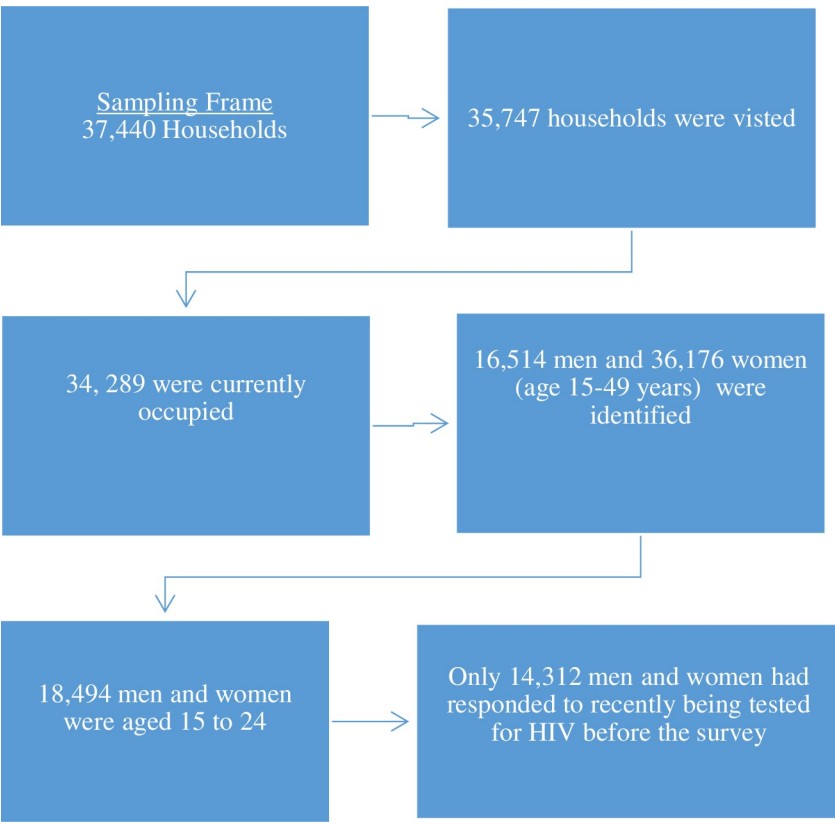

**Fig 1. Overview of the study sample.**

correctly. The first question probed whether they had ever heard of AIDS. The second question asks the respondents whether they could avoid the HIV/AIDS virus by having one uninfected partner. The third asks whether the respondents could get the HIV/AIDS virus through supernatural means. The fourth question asks the respondents whether they could avoid the HIV/AIDS virus by using a condom correctly every time. The fifth asks the respondents whether they could get the HIV/AIDS virus from mosquito bites. The sixth question asked whether the respondents could get the HIV virus by sharing food with a person who has HIV. The seventh asked the respondent whether a healthy-looking person may have AIDS virus. The range of scores was categorised as Low," "Moderate" and "High", with a score of 0–3 categorised as low knowledge, a score of 4–5 categorised as moderate knowledge and a score of 6–7 considered high knowledge.

Stigma belief is a misplaced fear of infection via everyday contact with those infected with HIV[35]. The dimension of internalised HIV stigma examined in this study is what is permitted by the five questions asked in the 5th MICS, all of which have dichotomous responses (Yes/No). The first question asks the respondents whether a female teacher with the HIV virus should be allowed to teach in school. The second asks whether the respondents would buy fresh vegetables from a shopkeeper living with HIV/AIDS. The third question poses whether the respondent would want it to remain a secret if a household member became infected with the HIV virus. The fourth asks the respondents whether they would be willing to care for a person with HIV/AIDS in the household. The fifth asks the respondents whether children with HIV should attend school. Exploratory factor analytic techniques suggest that these questions follow one underlying construct, with a Cronbach's Alpha test scale of 0.72. Stigma belief was

constructed and recoded as 1 "Extremely negative" 2 "Negative" 3 "Moderate" 4 "Positive" and 5 "Extremely positive".

The last critical independent variable, media exposure, is constructed from three variables on the frequency of exposure to media: reading print media, listening to the radio and watching television. The frequency of exposure was categorised as 1 "less than once a week" 2 "at least once a week" and 3 "almost every day". These were added up across all respondents to yield a total score of 0–9 per participant. The scores were recategorised as 1 "Low media exposure"(score of 0–3), 2 "Moderate media exposure" (score of 4–6) 3 "High media exposure" (score of 7–9) as used in other studies[4].

Household wealth was measured using the principal component analysis to surmise questions on types of roofing and general infrastructure in participants' home. The variable was already computed before making the data public. The contextual factors include geopolitical zones and residential area–classified as rural and urban.

## Statistical analysis

Sample weights were used in all analyses to adjust for disproportionate sampling and to obtain unbiased population parameter estimates. Descriptive statistics such as percentages and frequencies were presented to describe the characteristics of the study respondents. A list-wise logistic regression model was fitted to investigate the influence of demographic, behavioural, community/social and contextual factors on ever and recent testing of HIV. Two models were fitted to investigate the relationship succinctly. Model 1 is the baseline model that examines the association between each of the individual-level factors, household wealth and contextual factors and uptake of HIV testing. Model 2 is a multivariable logistic regression that includes demographic, behavioural, community/social and contextual factors. The regression results were interpreted using odds ratios (OR), with OR > 1 indicating a higher likelihood, OR = 1 showing no likelihood difference and OR < 1 indicating a lower probability. The level of significance was set at 0.05, while a confidence interval (CI) of 95% was used. Stata software (version 14) was used for all data analyses.

## Results

### Descriptive findings

The weighted social, demographic, contextual, and behavioural characteristics of the study respondents are presented in Table 1. More than half (57.7%) of the participants were between 15 to 19 years old, 64.6% lived in rural areas, 63.2% were female; more than two-thirds (79%) were single (never married), and 64.5% had secondary or technical education. Knowledge of HIV and transmission routes was mostly high or moderate among the participants, while some participants exhibited stigma beliefs related to people with HIV. Access to media was low among most of the participants (57.3%) and over half of the respondents have ever had sex (56.2%), but only 5.8% reported condom use.

### Bivariate findings

Tables 2 and 3 present the proportion of respondents who have tested for HIV at any time or recently across demographic, behavioural, community/social and contextual factors. As Table 2 shows, only a quarter of AYA had ever tested for HIV; however, the proportion varies by demographic, behavioural, social and contextual factors. More females than males (25.4% vs 20.8%), young adults than adolescents (33.1% vs 16.7%), tertiary-educated than no formal education (52.9% vs 10.4%), and urban than rural residents (27.2% vs 21.3%) had ever

**Table 1. Weighted sociodemographic, contextual and behavioural characteristics of study participants.**

| Characteristics | Frequency | Percent |
|---|---:|---:|
| Age group (in Years) | | |
| 15–19 | 8255 | 57.68 |
| 20–24 | 6057 | 42.32 |
| Gender | | |
| Female | 9049 | 63.23 |
| Male | 5263 | 36.77 |
| Marital Status | | |
| Never married | 11310 | 79.02 |
| Currently married | 2833 | 19.79 |
| Formerly married | 169 | 1.18 |
| Education | | |
| None | 2529 | 17.67 |
| Primary | 1237 | 8.64 |
| Secondary (& Tech) | 9226 | 64.46 |
| Higher | 1320 | 9.22 |
| Wealth Status | | |
| Poorest | 1807 | 12.63 |
| Second | 2580 | 18.03 |
| Middle | 3196 | 22.33 |
| Forth | 3272 | 22.86 |
| Richest | 3457 | 24.15 |
| Zone | | |
| North central | 3616 | 25.27 |
| North east | 3055 | 21.35 |
| North west | 3055 | 21.35 |
| South east | 1692 | 11.82 |
| South south | 2157 | 15.07 |
| South west | 1784 | 12.47 |
| Residence | | |
| Rural | 9251 | 64.64 |
| Urban | 5061 | 35.36 |
| Knowledge of HIV | | |
| Low | 956 | 6.68 |
| Moderate | 5197 | 36.31 |
| High | 8159 | 57.01 |
| Attitude towards HIV | | |
| Extremely Negative | 3234 | 22.60 |
| Negative | 2283 | 15.95 |
| Moderate | 2320 | 16.21 |
| Positive | 3025 | 21.14 |
| Extremely Positive | 3450 | 24.11 |
| Media Exposure | | |
| Low | 8199 | 57.29 |
| Moderate | 4612 | 32.22 |
| High | 1501 | 10.49 |
| Ever has sex | | |
| No | 8036 | 56.15 |

*(Continued)*

**Table 1.** (Continued)

| Characteristics | Frequency | Percent |
|---|---:|---:|
| Yes | 6276 | 43.85 |
| Condom Use | | |
| No | 13488 | 94.24 |
| Yes | 824 | 5.76 |
| Recently Tested HIV | | |
| No | 12370 | 86.43 |
| Yes | 1942 | 13.57 |

undergone HIV. Likewise, the rate of lifetime uptake of HIV testing was higher among participants who had ever engaged in sex (33.6%), exposed to media (39.9%), held positive attitude to people living with HIV (34.0%) and who had higher HIV knowledge (29.4%) compared to those who never had sex (16.5%%), never used condoms (22.0%), had low media exposure (18.3%), held extremely negative stigma belief (13.0%), and who had low knowledge of HIV (12.4%), respectively.

Only 13.6% of the respondents tested in the year preceding the survey and the proportion of those who recently tested varies by demographic, behavioural, social and contextual. The rate of recent HIV testing was highest among young adults aged 20–24 years, those who had higher education, resided in North Central Nigeria, had higher knowledge of HIV and exposure to media and those who held positive stigma belief about PLHIV (Table 3).

## Multivariable findings

Tables 4 and 5 summarise results of the multivariable models examining the association between the uptake of HIV testing and demographic, behavioural, social, and contextual factors. All demographic factors included in the study (age, gender, marital status, education and wealth) were significantly associated with having ever tested for HIV in the unadjusted regression model. The direction of the association persists for all demographic variables but the magnitude of the effect slightly reduced in the adjusted model. Young adults aged 20–24 (AOR 1.52, 95% CI 1.34–1.72) had higher odds of having been tested for HIV, compared to adolescents aged 15–19 years. Also, currently married young people were almost twice as likely to have ever tested for HIV compared to those who have never been married. Compared to respondents with no formal education, those with higher education (AOR 5.85, 95% CI 4.39–7.81) had a higher likelihood of having ever been tested for HIV. Moreover', young people in the richest wealth quintile were twice more likely to have ever been tested for HIV compared to those in the poorest wealth quintile.

Both place and geopolitical zone of residence were strongly associated with ever having been tested for HIV in the unadjusted regression model. The direction and magnitude of effect persist in the adjusted model. Adolescents and young adults residing in the north-central and south-south geopolitical zones had a higher likelihood of ever having been tested for HIV compared to those in the north-west.

Regarding the association between social factors and uptake of HIV testing, the unadjusted model shows that media exposure, knowledge of HIV, and stigma attitude were significantly related to lifetime uptake of HIV testing. The direction and magnitude of the effect remain consistent even in the adjusted model. Adolescents and young adults who had high HIV knowledge had higher odds of ever having been tested for HIV testing compared to those who had low HIV knowledge. Similarly, those who espoused a positive attitude towards PLHIV

**Table 2. Weighted proportions of adolescents who have ever tested for HIV by demographic, behavioural, social and contextual factors.**

| Characteristics (N = 14,312) | Yes (N) | (%) | 95% CI | P-Value |
|---|---|---|---|---|
| All participants | 3599 | 23.69 | [22.39,25.04] | |
| Age group (in Years) | | | | <0.001 |
| 15–19 | 1478 | 16.66 | [15.45,17.96] | |
| 20–24 | 2121 | 33.1 | [31.12,35.14] | |
| Gender | | | | <0.001 |
| Female | 2401 | 25.38 | [23.75,27.08] | |
| Male | 1198 | 20.76 | [19.13,22.5] | |
| Marital Status | | | | <0.001 |
| Never married | 2726 | 22.37 | [21.03,23.78] | |
| Currently married | 802 | 27.93 | [25.04,31.02] | |
| Formerly married | 71 | 33.23 | [24.96,42.67] | |
| Education | | | | <0.001 |
| None | 265 | 10.43 | [8.474,12.78] | |
| Primary | 233 | 19.02 | [15.87,22.63] | |
| Secondary (& Tech) | 2381 | 24.74 | [23.3,26.23] | |
| Higher | 720 | 52.89 | [49.18,56.57] | |
| Wealth Status | | | | <0.001 |
| Poorest | 194 | 9.738 | [7.761,12.15] | |
| Second | 501 | 17.88 | [15.47,20.57] | |
| Middle | 743 | 20.93 | [18.54,23.53] | |
| Forth | 970 | 28.48 | [26.01,31.09] | |
| Richest | 1191 | 33.7 | [31.49,35.99] | |
| Zone | | | | <0.001 |
| North west | 429 | 13.41 | [11.43,15.67] | |
| North central | 1118 | 34.77 | [31.57,38.1] | |
| North east | 374 | 16.52 | [12.9,20.92] | |
| South east | 481 | 30.03 | [27.24,32.99] | |
| South south | 764 | 36.92 | [33.76,40.19] | |
| South west | 433 | 26.6 | [23.75,29.66] | |
| Residence | | | | <0.001 |
| Rural | 2145 | 21.34 | [19.76,23] | |
| Urban | 1454 | 27.16 | [24.85,29.6] | |
| Ever has sex | | | | <0.001 |
| No | 1429 | 16.47 | [15.15,17.88] | |
| Yes | 2170 | 33.59 | [31.62,35.63] | |
| Condom Use | | | | <0.001 |
| No | 3132 | 22.02 | [20.73,23.37] | |
| Yes | 467 | 55.27 | [50.62,59.83] | |
| Knowledge of HIV | | | | <0.001 |
| Low | 137 | 12.39 | [10.13,15.08] | |
| Moderate | 985 | 16.43 | [15.07,17.89] | |
| High | 2477 | 29.44 | [27.62,31.32] | |
| Attitude towards HIV | | | | <0.001 |
| Extremely Negative | 460 | 13.04 | [11.52,14.74] | |
| Negative | 450 | 18.75 | [16.8,20.86] | |
| Moderate | 552 | 22.14 | [20.07,24.36] | |
| Positive | 872 | 27.41 | [25.17,29.77] | |

*(Continued)*

**Table 2.** (Continued)

| Characteristics (N = 14,312) | Yes (N) | (%) | 95% CI | P-Value |
|---|---|---|---|---|
| Extremely Positive | 1265 | 33.97 | [30.97,37.1] | |
| Media Exposure | | | | <0.001 |
| Low | 1616 | 18.28 | [16.8,19.87] | |
| Moderate | 1380 | 28.13 | [26.08,30.28] | |
| High | 603 | 39.85 | [36.41,43.4] | |

Pearson Chi-squared test was used to obtain P-values. This applies for all P-values

were three times more likely to have ever tested for HIV compared to those with an extremely negative attitude towards PLHIV. Additionally, individuals frequently exposed to the media had a higher likelihood of having ever tested for HIV.

Adolescents' and young adults' behavioural characteristics, such as having engaged in sex and condom use, were significantly associated with ever having tested for HIV in the unadjusted model. The strength of the association reduced in the adjusted model, but the direction of the effect persists. Young people who have ever had sex or used condoms had higher odds of having ever tested for HIV.

The relationship between recent uptake of HIV testing and demographic, social, contextual, and behavioural factors was examined using adjusted, and unadjusted logistic regression models and the results are presented in Table 5. Of all demographic factors, only age, educational level and wealth status were significantly positively related to recent uptake of HIV testing in the unadjusted model. However, the magnitude of the effect of these variables reduced substantially in the adjusted model, except for currently married where the effect size increased. Currently married adolescents and young adults had a higher likelihood of recent uptake of HIV testing compared to their never-married counterparts. Similarly, young adults aged 20 to 24 were 20% more likely to have recently tested for HIV compared to adolescents aged 15 to 19. Also, AYA who had higher education were five times likely to have recently tested for HIV compared to those who had no formal education.

Place and geopolitical zones were positively associated with recent uptake of HIV testing in the unadjusted model. The magnitude of effect reduced in the adjusted model for both place of residence and geopolitical zones, but the direction of effect persists only for geopolitical zones. Favourable attitude towards PLHIV, high HIV knowledge, and media exposure were associated with higher odds of recent uptake of HIV testing. The direction of effect persists in the adjusted model, but the magnitude of the effect reduced. Ever had sex and condom use were associated with a higher likelihood of recent uptake of HIV testing and the magnitude and direction of effects persist in the adjusted regression model.

## Discussion

This study examined the coverage and factors associated with HIV testing uptake among AYA in Nigeria. Our findings show that HIV testing uptake is low among young people aged 15–24 years. The rate of HIV testing found in this study shows only a minimal increase in the uptake of HIV testing in this age group when compared to the NDHS 2013[4]. While there is evidence that HIV testing uptake has increased significantly among pregnant women [37], it appears that such improvement is yet to be achieved with young people. Interventions to increase HIV testing uptake seem to focus on pregnant women [37], neglecting young people who are among the most at-risk of contracting HIV [9]. Given the importance of early diagnosis to the

**Table 3. Weighted proportions of adolescents recently tested for HIV by demographic, behavioural, social and contextual factors.**

| Characteristics N = 14,312 | Yes (N) | (%) | 95% CI | P-value |
|---|---|---|---|---|
| All participants | 1942 | 12.42 | [11.49,13.41] | |
| Age group (in Years) | | | | <0.001 |
| 15–19 | 865 | 9.47 | [8.57,10.46] | |
| 20–24 | 1077 | 16.36 | [15.01,17.79] | |
| Gender | | | | 0.37 |
| Female | 1240 | 12.69 | [11.61,13.85] | |
| Male | 702 | 11.95 | [10.61,13.42] | |
| Marital Status | | | | 0.675 |
| Never married | 1534 | 12.47 | [11.43,13.6] | |
| Formerly married | 373 | 12.09 | [10.44,13.96] | |
| Currently married | 35 | 14.78 | [10.01,21.3] | |
| Education | | | | <0.001 |
| None | 109 | 4.08 | [3.02,5.48] | |
| Primary | 121 | 8.86 | [7.09,11.02] | |
| Secondary (& Tech) | 1312 | 13.52 | [12.39,14.74] | |
| Higher | 400 | 28.46 | [25.44,31.69] | |
| Wealth Status | | | | <0.001 |
| Poorest | 107 | 5.63 | [4.23,7.45] | |
| Second | 270 | 8.71 | [7.16,10.55] | |
| Middle | 398 | 10.94 | [9.35,12.76] | |
| Forth | 502 | 14.56 | [12.87,16.42] | |
| Richest | 665 | 18.21 | [16.41,20.17] | |
| Zone | | | | <0.001 |
| North west | 209 | 5.89 | [4.869,7.112] | |
| North central | 670 | 20.90 | [18.07,24.04] | |
| North east | 204 | 7.20 | [5.63,9.16] | |
| South east | 199 | 12.73 | [10.6,15.21] | |
| South south | 441 | 22.88 | [19.98,26.07] | |
| South west | 219 | 13.63 | [11.47,16.12] | |
| Residence | | | | 0.01 |
| Rural | 1150 | 11.30 | [10.23,12.47] | |
| Urban | 792 | 14.05 | [12.34,15.97] | |
| Ever has sex | | | | <0.001 |
| No | 813 | 8.98 | [8.043,10] | |
| Yes | 1129 | 17.13 | [15.74,18.62] | |
| Condom Use | | | | <0.001 |
| No | 1673 | 11.35 | [10.48,12.3] | |
| Yes | 269 | 32.50 | [28.26,37.05] | |
| Knowledge of HIV | | | | <0.001 |
| Low | 74 | 6.72 | [5.16,8.72] | |
| Moderate | 537 | 8.70 | [7.75,9.765] | |
| High | 1331 | 15.35 | [14.04,16.75] | |
| Attitude towards HIV | | | | <0.001 |
| Extremely Negative | 210 | 6.22 | [5.21,7.41] | |
| Negative | 202 | 8.27 | [7.06,9.66] | |
| Moderate | 262 | 10.57 | [9.10,12.24] | |
| Positive | 500 | 15.10 | [13.39,16.97] | |

*(Continued)*

**Table 3.** (Continued)

| Characteristics N = 14,312 | Yes (N) | (%) | 95% CI | P-value |
|---|---|---|---|---|
| Extremely Positive | 768 | 19.34 | [16.96,21.97] | |
| Media Exposure | | | | <0.001 |
| Low | 829 | 9.02 | [8.02,10.12] | |
| Moderate | 767 | 15.30 | [13.82,16.91] | |
| High | 346 | 22.31 | [19.53,25.36] | |

Pearson Chi-squared test was used to obtain p -values. This applies for all p values

treatment outcomes of PLHIV [10, 11], there is a need to develop a policy framework targeted at expanding access and uptake of HIV testing among young people.

Consistent with previous studies [15, 17, 19], our study shows that demographic factors, such as age, sex, marital status, and education level, are significantly associated with uptake of HIV testing. Uptake of HIV testing tends to increase with increasing age. Also, females in general and married young girls/women in particular, have higher odds of HIV testing compared to males in Nigeria. Adolescents and young women who had attended antenatal care or given birth have more opportunities of getting tested compared to young men who may have limited reasons to visit the clinic. Although the 'opt-out' strategy is the main policy thrust to increase HIV testing uptake in Nigeria [38], its implementation appears to have targeted mainly pregnant women and neglected other categories of patients attending clinics for other reasons besides pregnancy.

Adolescents and young adults educated up to post-secondary school level had a higher odds of uptake of HIV testing compared to those not educated or with only primary level of education. The importance of education for improved health outcomes has been demonstrated in previous studies [39, 40]. Educated young adults are more likely to have good knowledge of HIV and better able to assess their risk. Knowledge of HIV and risk perceptions have been established to be associated with uptake of HIV testing [5]. Investment in the education of young people will not only increase the uptake of HIV testing but also improve other health outcomes.f

Another important finding of this study is that place and region of residence are positively associated with uptake of HIV testing. The distribution of HIV testing appears to mirror HIV prevalence by state and indicates areas where interventions are concentrated (See S1 Table). While it is important to prioritise areas needing urgent intervention, it is also critical that regions of low prevalence are not neglected.

Consistent with previous studies and behavioural theories, HIV knowledge, stigma belief, and media exposure were associated with uptake of HIV testing[4, 5, 20]. Individuals with higher HIV knowledge were more likely to have ever been tested for HIV compared to those with low knowledge of HIV, indicating the need for sustaining efforts directed at educating people about HIV and the benefits of HIV testing. Likewise, media exposure was associated with higher odds of HIV testing uptake, which further reinforces the role of media towards raising awareness about HIV, the benefits of HIV testing and treatment, and communicating recent advances like availability of pre-exposure prophylaxis and more importantly the "Undetectable Equals Untransmittable" message. Nevertheless, stigma belief about HIV remains associated with lower odds of HIV testing uptake, suggesting that stigma remains a significant barrier in the Nigerian context. While it is unsurprising that stigma belief about HIV is associated with low uptake of HIV testing, the extent of stigma belief expressed by young people in Nigeria, no doubt, indicate misguided knowledge or gaps in knowledge of HIV. Thus efforts

**Table 4. Multivariable logistic regression analysis showing predictors of uptake of HIV testing in the respondent's lifetime.**

| Characteristics (N = 14,312) | Unadjusted Odds ratio [95% CI] | | Adjusted Odds ratio [95% CI] | |
|---|---|---|---|---|
| Age group | | | | |
| 15–19 | Ref | | | |
| 20–24 | 2.47 | [2.24,2.73] *** | 1.52 | [1.34,1.72] *** |
| Gender | | | | |
| Female | Ref | | | |
| Male | 0.77 | [0.70,0.85] *** | 0.95 | [0.84,1.08] |
| Marital Status | | | | |
| Never married | Ref | | | |
| Currently married | 1.34 | [1.20,1.51] *** | 2.42 | [1.98,2.97] *** |
| Formerly married | 1.73 | [1.17,2.56] *** | 1.54 | [0.89,2.67] |
| Education | | | | |
| None | Ref | | | |
| Primary | 2.02 | [1.58,2.57] *** | 1.83 | [1.39,2.39] *** |
| Secondary (& Tech) | 2.82 | [2.38,3.35] *** | 2.68 | [2.12,3.38] *** |
| Higher | 9.64 | [7.82,11.87] *** | 5.85 | [4.39,7.81] *** |
| Wealth Status | | | | |
| Poorest | Ref | | | |
| Second | 2.02 | [1.62,2.52] *** | 1.57 | [1.24,1.99] *** |
| Middle | 2.45 | [2.00,3.012] *** | 1.48 | [1.16,1.87] *** |
| Forth | 3.69 | [3.01,4.53] *** | 1.85 | [1.43,2.39] *** |
| Richest | 4.71 | [3.87,5.73] *** | 1.99 | [1.53,2.60] *** |
| Zone | | | | |
| Northwest | Ref | | | |
| North central | 3.44 | [2.97,3.99]*** | 3.38 | [2.87,3.97] *** |
| North east | 1.28 | [1.04,1.58] ** | 1.29 | [1.03,1.61] ** |
| South east | 2.77 | [2.35,3.28] *** | 2.13 | [1.75,2.58] *** |
| South south | 3.78 | [3.24,4.41] *** | 2.55 | [2.11,3.07] *** |
| South west | 2.34 | [1.97,2.78] *** | 1.50 | [1.22,1.85] *** |
| Residence | | | | |
| Rural | Ref | | | |
| Urban | 1.37 | [1.25,1.52] *** | 1.00 | [0.86,1.16] |
| Knowledge of HIV | | | | |
| Low | Ref | | | |
| Moderate | 1.39 | [1.10,1.75] *** | 1.11 | [0.85,1.44] |
| High | 2.95 | [2.36,3.69] *** | 1.62 | [1.24,2.11]*** |
| Attitude towards HIV | | | | |
| Extremely Negative | Ref | | | |
| Negative | 1.54 | [1.29,1.83] *** | 1.34 | [1.11,1.62] *** |
| Moderate | 1.9 | [1.60,2.24] *** | 1.55 | [1.28,1.87] *** |
| Positive | 2.52 | [2.16,2.94] *** | 2.03 | [1.70,2.43] *** |
| Extremely Positive | 3.43 | [2.94,4.00] *** | 2.93 | [2.47,3.49]*** |
| Media Exposure | | | | |
| Low | Ref | | | |
| Moderate | 1.75 | [1.57,1.95] *** | 1.17 | [1.02,1.34] *** |
| High | 2.96 | [2.56,3.42] *** | 1.64 | [1.36,1.97] *** |
| Ever has sex | | | | |
| No | Ref | | | |

*(Continued)*

**Table 4.** (Continued)

| Characteristics (N = 14,312) | Unadjusted Odds ratio [95% CI] | | Adjusted Odds ratio [95% CI] | |
|---|---|---|---|---|
| Yes | 2.57 | [2.32,2.83] *** | 1.72 | [1.49,1.98] *** |
| Condom Use | | | | |
| No | Ref | | | |
| Yes | 4.37 | [3.68,5.20] *** | 1.84 | [1.49,2.28] *** |

***P values <0.001

*P values <0.01

Ref indicates the baseline category (reference) for each variable.

should be placed on improving knowledge of young people about HIV in Nigeria to improve health outcomes at the population level.

Besides the individual level barriers, structural barrier also limits AYA' access to HIV testing. For example, the legal age of consent for independent HIV testing is 18 years in Nigeria [41]. This law presents a structural barrier to accessing HIV testing services [42], as young people will mostly likely not bring their parents to provide consent and also providers would want to follow the law. The experts in the field of sexual and reproductive health (SRH) in Nigeria, during a strategic engagement held with the federal ministry of health, recommended age 14 yrs for the age of consent to HIV testing. We consider removing this legal barrier, as recommended by experts in the field of SRH in Nigeria, to be critical for expanding access to HIV testing in Nigeria. The low level of HIV testing among AYA in Nigeria has severe implication for the spread of HIV, cost of treatment and HIV related-deaths. Nigerian's AYA are disproportionately affected by HIV and should be a priority group of HIV interventions. Expanding access to HIV testing for AYA remains a priority intervention to achieve the country's 2020 target of 95% testing rate. Improving young people's knowledge of HIV perhaps through the media as well as other interventions such as home-based testing and the "opt-out" strategy will be critical to substantially increase HIV testing uptake among young people in Nigeria.

### Study strengths and limitations

The findings reported in this study have some limitations. First, HIV testing is based on self-reporting and while this is a major event that participants should recall, social desirability bias cannot be ruled out even though its impact on the findings should be minimal. Second, the reported association should not be interpreted as causation since the data is cross-sectional, and it is difficult to establish which event is a precondition for another. Nevertheless, the findings of the study are reassuring given that there are based on nationally representative data. The study provides much-needed data for strategic planning and programming to improve health outcomes among young people in the country.

### Conclusion

We examined the coverage and associated factors of HIV testing uptake among young people in Nigeria using nationally representative data. Our study found a low rate of HIV testing that varies by demographic, behavioural, social, and contextual characteristics. The findings indicate a slight increase in the rate of HIV testing among young people in the country. However, the rate of increase is too slow and fell far short of the standard required to achieve the

**Table 5. Multivariable logistic regression analysis showing predictors of recent uptake of HIV testing.**

| Characteristics (N = 14,312) | Unadjusted Odds ratio [95% CI] | | Adjusted Odds ratio [95% CI] | |
|---|---|---|---|---|
| Age group | | | | |
| 15–19 | Ref | | | |
| 20–24 | 1.87 | [1.66,2.10]*** | 1.20 | [1.03,1.40]*** |
| Gender | | | | |
| Female | Ref | | | |
| Male | 0.93 | [0.83,1.05] | 1.09 | [0.94,1.25] |
| Marital Status | | | | |
| Never married | Ref | | | |
| Currently married | 0.97 | [0.83,1.12] | 1.67 | [1.33,2.091]*** |
| Formerly married | 1.22 | [0.79,1.88] | 1.09 | [0.66,1.81] |
| Education | | | | |
| None | Ref | | | |
| Primary | 2.29 | [1.66,3.16]*** | 1.81 | [1.29,2.55]*** |
| Secondary (& Tech) | 3.68 | [2.89,4.67]*** | 2.78 | [2.08,3.72]*** |
| Higher | 9.36 | [7.14,12.26]*** | 4.77 | [3.39,6.71]*** |
| Wealth Status | | | | |
| Poorest | Ref | | | |
| Second | 1.60 | [1.22,2.10]*** | 1.11 | [0.84,1.47] |
| Middle | 2.06 | [1.58,2.68]*** | 1.05 | [0.78,1.42] |
| Forth | 2.86 | [2.21,3.70]*** | 1.15 | [0.85,1.57] |
| Richest | 3.73 | [2.91,4.80]*** | 1.26 | [0.91,1.75] |
| Zone | | | | |
| Northwest | Ref | | | |
| North central | 4.22 | [3.48,5.12]*** | 3.61 | [2.95,4.43]*** |
| North east | 1.24 | [0.96,1.59] | 1.20 | [0.92,1.56] |
| South east | 2.33 | [1.85,2.94]*** | 1.60 | [1.25,2.06]*** |
| South south | 4.74 | [3.87,5.81]*** | 3.04 | [2.42,3.82]*** |
| South west | 2.52 | [2.00,3.17]*** | 1.70 | [1.31,2.21]*** |
| Residence | | | | |
| Rural | Ref | | | |
| Urban | 1.28 | [1.14,1.45]*** | 0.96 | [0.82,1.13] |
| Knowledge of HIV | | | | |
| Low | Ref | | | |
| Moderate | 1.32 | [0.99,1.77] | 1.01 | [0.74,1.37] |
| High | 2.51 | [1.90,3.33]*** | 1.24 | [0.91,1.69] |
| Attitude towards HIV | | | | |
| Extremely Negative | Ref | | | |
| Negative | 1.36 | [1.07,1.72]** | 1.16 | [0.91,1.48] |
| Moderate | 1.78 | [1.42,2.23]*** | 1.43 | [1.13,1.81]*** |
| Positive | 2.68 | [2.18,3.29]*** | 2.15 | [1.73,2.68]*** |
| Extremely Positive | 3.62 | [2.97,4.40]*** | 3.04 | [2.47,3.75]*** |
| Media Exposure | | | | |
| Low | Ref | | | |
| Moderate | 1.82 | [1.60,2.07]*** | 1.24 | [1.07,1.45]*** |
| High | 2.90 | [2.45,3.43]*** | 1.63 | [1.33,2.00]*** |
| Ever has sex | | | | |
| No | Ref | | | |

*(Continued)*

**Table 5.** (Continued)

| Characteristics (N = 14,312) | Unadjusted Odds ratio [95% CI] | | Adjusted Odds ratio [95% CI] | |
|---|---|---|---|---|
| Yes | 2.10 | [1.86,2.36]*** | 1.62 | [1.37,1.92]*** |
| Condom Use | | | | |
| No | Ref | | | |
| Yes | 3.76 | [3.13,4.52]*** | 1.78 | [1.41,2.24]*** |

***P values <0.001

*P values <0.01

Ref indicates the baseline category (reference) for each variable.

UNAIDS first 95 and the national goal of testing 95% of people in the country by 2020. Improving young people's knowledge of HIV perhaps through the media as well as other interventions such as home-based testing and the "opt-out" strategy will be critical to substantially increase HIV testing uptake among young people in Nigeria.

## Supporting information

**S1 Table. HIV testing by states.**
(DOCX)

## Author Contributions

**Conceptualization:** Anthony Idowu Ajayi, Oluwafemi Emmanuel Awopegba, Oluwafemi Atanda Adeagbo, Boniface Ayanbekongshie Ushie.

**Data curation:** Anthony Idowu Ajayi, Oluwafemi Emmanuel Awopegba.

**Formal analysis:** Anthony Idowu Ajayi, Oluwafemi Emmanuel Awopegba.

**Methodology:** Anthony Idowu Ajayi, Oluwafemi Emmanuel Awopegba, Oluwafemi Atanda Adeagbo, Boniface Ayanbekongshie Ushie.

**Supervision:** Boniface Ayanbekongshie Ushie.

**Writing – original draft:** Anthony Idowu Ajayi, Oluwafemi Emmanuel Awopegba, Oluwafemi Atanda Adeagbo, Boniface Ayanbekongshie Ushie.

**Writing – review & editing:** Anthony Idowu Ajayi, Oluwafemi Emmanuel Awopegba, Oluwafemi Atanda Adeagbo, Boniface Ayanbekongshie Ushie.

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
