## [Decision Letter · Decision Letter 0]

17 Apr 2020

PONE-D-20-04813

Low coverage of HIV testing among adolescents and young adults in Nigeria: implication for achieving the UNAIDS first 90

PLOS ONE

Dear Dr Ajayi,

Thank you for submitting your manuscript to PLOS ONE. After careful consideration, we feel that it has merit but does not fully meet PLOS ONE’s publication criteria as it currently stands. Therefore, we invite you to submit a revised version of the manuscript that addresses the points raised during the review process.

We would appreciate receiving your revised manuscript by Jun 01 2020 11:59PM. To enhance the reproducibility of your results, we recommend that if applicable you deposit your laboratory protocols in protocols.io, where a protocol can be assigned its own identifier (DOI) such that it can be cited independently in the future. For instructions see: http://journals.plos.org/plosone/s/submission-guidelines#loc-laboratory-protocols

We look forward to receiving your revised manuscript.

Kind regards,

Olanrewaju Oladimeji 

Academic Editor

PLOS ONE

Additional Editor Comments (if provided):

In addition to the reviewers comments, the authors should keep the flowchart vertically downward and easure that the references are in vancouver square bracket.

2. Please ensure that you refer to Figure 1 in your text as, if accepted, production will need this reference to link the reader to the figure.

3. We note you have included a table to which you do not refer in the text of your manuscript. Please ensure that you refer to Table 4 in your text; if accepted, production will need this reference to link the reader to the Table.

Reviewers' comments:

Reviewer's Responses to Questions

**Comments to the Author**

1. Is the manuscript technically sound, and do the data support the conclusions?

Reviewer #1: Yes

Reviewer #2: Yes

2. Has the statistical analysis been performed appropriately and rigorously? 

Reviewer #1: Yes

Reviewer #2: Yes

3. Have the authors made all data underlying the findings in their manuscript fully available?

Reviewer #1: Yes

Reviewer #2: Yes

4. Is the manuscript presented in an intelligible fashion and written in standard English?

Reviewer #1: Yes

Reviewer #2: No

5. Review Comments to the Author

Reviewer #1: This article is a secondary analysis is publicly available data and establishes low HIV testing coverage among adolescent and young adults (AYA). The article is well written with logical presentation of its findings.

However, the bivarate analysis on table 2 seems not to add up. For instance, there were a total of 14312 eligible AYA. Of these, 3599 responded to ever tested for HIV which gives 25.1% (i.e. 3599/14312) but the tables shows "23.69%". The same is true for all the variables on table 2. Though minor, it would be helpful to either explain why there is a variation or correct the error if that is the case.

Secondly, table 5 show the multivariate logistics regression, is there any reason for adding variables that do not show any bivariate association (e.g. "knowledge of HIV; marital status") in the final model?

Reviewer #2: This paper presents an analysis of MICS data with respect to HIV testing coverage and associated factors among 15 to 24 yr olds in Nigeria. It is a timely analysis for Nigeria in the transition from 90-90-90 UNAIDS targets to 95-95-95 targets.

GENERAL:

1. Please check the tense in the Methods and Results. Best to use past tense for what was done and the results obtained.

2. Add "years" whenever age is mentioned eg 15-24 years, and not simply 15-24.

ABSTRACT:

1. Note and correct the transposition of MICS: Multiple Indicator Cluster Survey (not MCIS)

2. Would be important to add gender info to Abstract results ie stats on higher testing rates among girls than boys

3. "The HIV testing coverage among AYA in Nigeria is well below the national target of 95% testing indicated in the national HIV/AIDS strategic framework (2017-2021)." Suggest delete the word "testing" after "95%"

4. "reducing negative stigma belief through the media campaign". Suggest delete the word "the" and add "s" to campaign.

INTRO:The info is fairly well-written but noticeably devoid of actual epi numbers and the source refs.

1. The first, and especially second para narrate epi info but do not actually state the numbers. Please state and reference the actual epi numbers so that the reader is well-informed. At least, state and reference the actual data for sub-Saharan Africa.

2. Suggest: "AYA aged 15-24 years bear a disproportionate burden of HIV, particularly for new infections and AIDS-related deaths. In sub-Saharan Africa," ...State the proportions for SSA.

3. "Also, they are more unlikely (*suggest less likely*) to have tested for HIV, have the lowest level of adherence to ART and viral load suppression." As I stated above, please provide the actual numbers on likelihood of HIV testing, adherence, and VL suppression. Please do that for the rest of the Intro section.

4. Define WHO acronym at first use; no need to use the acronym since it is used only once in the paper. Similar for SSA...no need for the acronym; used only twice.

5. I strongly suggest replacing 90-90-90 definition with 95-95-95 and providing a reference for the definition. We are in 2020 and 90-90-90 is no longer relevant.

6. ...couples' (apostrophe added) open communication about HIV, expanding access to HIV testing and treatment, *as* well as having a social support network...

7. Suggest: *At 1.5%*, Nigeria has a low prevalence of HIV, but the country’s large population *of >200 million* (ref) means that the number of people living with HIV in the country is *substantial*.

8. *A significant proportion; up to xx%* of new infections occurs among adolescents and young adults in the country." Provide actual epi data and reference.

9. "Most studies on HIV testing among young people in Nigeria are not nationally representative or

are now *outdated* 4,5,16,26"

10. "As such, recent nationally representative data, such as the Multiple Indicator (no "s")

Cluster Survey (MICS) *(suggest capitalize first letters)*, could help assess current HIV testing rates among young people and progress recorded since the scaling *up* of such services in Nigeria."

11. Health Belief Model (capitalize first letters please)

12. Suggest deleting this sentence. Other adjacent statements send the same message in more formal language. "People can test for HIV knowing they could lead a normal and healthy life."

MATERIALS AND METHODS

1. Use the term fifth or 5th (superscript th)

2. Use past tense in Methods, including in Ethical Statement: We *did* not need...

3. Study Population:suggest "There were 52,690 MICS respondents (xx male and xx female). However, the focus of this study was AYA 15-24 yrs, who comprised 18, xxx (xx% out of all respondents), who met selection criteria. The analysis was further limited to 14, xxx AYA who responded to the questions on HIV testing".

4. "Our dependent variables of interest in this study are two two-category nominal measure of uptake

of HIV testing." Is the repeat of the word "two" a typo?

5. The demographic variable should be "formerly married" (not formally). Also correct that in the Tables.

6. Use quotation marks if variables are not in brackets: "The behavioural variables are "ever had sex" and "condom use".

7. The first question probed whether they *had* ever heard of AIDS.

8. with a score of 0-3 Categorised as low knowledge (use small c)

9. I believe term is household wealth, not house wealth? "The variable was already computed (delete "as")

before making the data public." "Model 2", not "Models 2"

10. "...who had low knowledge of HIV (12.4%%)"-delete second %

RESULTS

1. Again, please check your tenses and use past tense

2. it is also critical that regions of low *prevalence* are not neglected

3. (UDOH & USHIE, 2019)-please format this ref

4. "Undetectable Equals Untransmittable" best to write in quotes and capitalize first letters. It's a brand strategic message.

DISCUSSION

1. Study strengths and limitations: "the data is a cross-sectional" -delete "a"

2. Conclusions: The findings indicate a slight increase in the rate of HIV testing among young people in the country but the rate of increase however too slow and fell far short of the standard required to achieve the UNAIDS first 90 and the national goal of testing 95% of people in the country by 2020.-this sentence is too long. Please break it up for easier reading and comprehension.

3. A significant concern for this section is the lack of mention of age of consent and legal barriers to access to testing. Nigeria still has an applied age of consent to HIV testing of 18 years. There is however a 2014 National Consensus document that recommends age 14 yrs for the lower age of consent (Ref: Guidelines for Young Persons' Participation in Research and Access to Sexual and Reproductive Health Services in Nigeria). I strongly recommend a short narrative on this issue, making reference to the Child Rights Act, the 2014 consensus as above, some of the articles by Prof Morenike Folayan and colleagues on adolescent rights and access to SRH services, and the recent Bulletin of the WHO publication by McKinnon and Vandermorris: National age-of-consent laws and adolescent HIV testing in sub-Saharan Africa: a propensity-score matched study.

6. PLOS authors have the option to publish the peer review history of their article (what does this mean?). If published, this will include your full peer review and any attached files.

Reviewer #1: Yes: Okpokoro Evaezi

Reviewer #2: No

---

## [Author Response · Author response to Decision Letter 0]

30 Apr 2020

Reviewer #1: This article is a secondary analysis is publicly available data and establishes low HIV testing coverage among adolescent and young adults (AYA). The article is well written with logical presentation of its findings.

Response : we are thankful for the positive feedback.

However, the bivarate analysis on table 2 seems not to add up. For instance, there were a total of 14312 eligible AYA. Of these, 3599 responded to ever tested for HIV which gives 25.1% (i.e. 3599/14312) but the tables shows "23.69%". The same is true for all the variables on table 2. Though minor, it would be helpful to either explain why there is a variation or correct the error if that is the case.

Response: This is not an error. The percentages presented are from the weighted samples. The 23.69% is inaccurate given the complex sampling adopted. The weighted results are the accurate results. We have retitled the tables to reflect that the percentages are weighted.

Secondly, table 5 show the multivariate logistics regression, is there any reason for adding variables that do not show any bivariate association (e.g. "knowledge of HIV; marital status") in the final model?

Response: The best practice is to include variables in a model based on extant literature and theories (1).Zellner D, Keller F, Zellner GE. Variable selection in logistic regression models. Communications in Statistics-Simulation and Computation. 2004 Jan 2;33(3):787-805. 2) Peter Flom. 2018. Stopping stepwise: Why stepwise selection is bad and what you should use instead 3) https://personal.utdallas.edu/~pkc022000/6390/SP06/NOTES/Logistic_Regression_4.pdf). 

The Health Belief Model and previous studies informed our selection of variables to include in our model. 

Reviewer #2: This paper presents an analysis of MICS data with respect to HIV testing coverage and associated factors among 15 to 24 yr olds in Nigeria. It is a timely analysis for Nigeria in the transition from 90-90-90 UNAIDS targets to 95-95-95 targets.

Response: many thanks for the positive feedback

GENERAL:

1. Please check the tense in the Methods and Results. Best to use past tense for what was done and the results obtained.

Response: We have checked the tenses and reported in past tenses.

2. Add "years" whenever age is mentioned eg 15-24 years, and not simply 15-24.

Response: Done

ABSTRACT:

1. Note and correct the transposition of MICS: Multiple Indicator Cluster Survey (not MCIS)

Response: Many thanks for correction. We have implemented it. 

2. Would be important to add gender info to Abstract results ie stats on higher testing rates among girls than boys

Response: we have added a sentence on gender difference in HIV testing.

3. "The HIV testing coverage among AYA in Nigeria is well below the national target of 95% testing indicated in the national HIV/AIDS strategic framework (2017-2021)." Suggest delete the word "testing" after "95%"

Response: deleted

4. "reducing negative stigma belief through the media campaign". Suggest delete the word "the" and add "s" to campaign.

Response. Done

INTRO:The info is fairly well-written but noticeably devoid of actual epi numbers and the source refs.

1. The first, and especially second para narrate epi info but do not actually state the numbers. Please state and reference the actual epi numbers so that the reader is well-informed. At least, state and reference the actual data for sub-Saharan Africa.

2. Suggest: "AYA aged 15-24 years bear a disproportionate burden of HIV, particularly for new infections and AIDS-related deaths. In sub-Saharan Africa," ...State the proportions for SSA.

Response: we have included the actual data by following this suggestion

3. "Also, they are more unlikely (*suggest less likely*) to have tested for HIV, have the lowest level of adherence to ART and viral load suppression." As I stated above, please provide the actual numbers on likelihood of HIV testing, adherence, and VL suppression. Please do that for the rest of the Intro section.

Response: We have included the actual data by following this suggestion

4. Define WHO acronym at first use; no need to use the acronym since it is used only once in the paper. Similar for SSA...no need for the acronym; used only twice.

Response: Done

5. I strongly suggest replacing 90-90-90 definition with 95-95-95 and providing a reference for the definition. We are in 2020 and 90-90-90 is no longer relevant.

Response: Done

6. ...couples' (apostrophe added) open communication about HIV, expanding access to HIV testing and treatment, *as* well as having a social support network...

Response: corrected

7. Suggest: *At 1.5%*, Nigeria has a low prevalence of HIV, but the country’s large population *of >200 million* (ref) means that the number of people living with HIV in the country is *substantial*.

Response: Many thanks for this suggestion. We have implemented the change.

8. *A significant proportion; up to xx%* of new infections occurs among adolescents and young adults in the country." Provide actual epi data and reference.

Response: We have added the data (up to 34%), 41000 out of 120,000 new infections.

9. "Most studies on HIV testing among young people in Nigeria are not nationally representative or

are now *outdated* 4,5,16,26"

Response: Many thanks for the suggestion. We have implemented the suggestion.

10. "As such, recent nationally representative data, such as the Multiple Indicator (no "s")

Cluster Survey (MICS) *(suggest capitalize first letters)*, could help assess current HIV testing rates among young people and progress recorded since the scaling *up* of such services in Nigeria."

Response: this has been corrected

11. Health Belief Model (capitalize first letters please)

Response: done 

12. Suggest deleting this sentence. Other adjacent statements send the same message in more formal language. "People can test for HIV knowing they could lead a normal and healthy life."

Response: done

MATERIALS AND METHODS

1. Use the term fifth or 5th (superscript th)

Response: we have changed 5th to fifth

2. Use past tense in Methods, including in Ethical Statement: We *did* not need...

Response: done

3. Study Population:suggest "There were 52,690 MICS respondents (xx male and xx female). However, the focus of this study was AYA 15-24 yrs, who comprised 18, xxx (xx% out of all respondents), who met selection criteria. The analysis was further limited to 14, xxx AYA who responded to the questions on HIV testing".

Response: many thanks for this suggestion. We have revised accordingly. 

4. "Our dependent variables of interest in this study are two two-category nominal measure of uptake

of HIV testing." Is the repeat of the word "two" a typo?

Response: typo has been corrected 

5. The demographic variable should be "formerly married" (not formally). Also correct that in the Tables.

Response: we have made the correction throughout the manuscript. 

6. Use quotation marks if variables are not in brackets: "The behavioural variables are "ever had sex" and "condom use".

Response: done

7. The first question probed whether they *had* ever heard of AIDS.

Response: corrected

8. with a score of 0-3 Categorised as low knowledge (use small c)

Response: corrected

9. I believe term is household wealth, not house wealth? "The variable was already computed (delete "as")before making the data public." "Model 2", not "Models 2"

Response: done

10. "...who had low knowledge of HIV (12.4%%)"-delete second %

Response: done

RESULTS

1. Again, please check your tenses and use past tense

Response: done

2. it is also critical that regions of low *prevalence* are not neglected

Response: corrected

3. (UDOH & USHIE, 2019)-please format this ref

Response: done

4. "Undetectable Equals Untransmittable" best to write in quotes and capitalize first letters. It's a brand strategic message.

Response: done

DISCUSSION

1. Study strengths and limitations: "the data is a cross-sectional" -delete "a"

Response: corrected

2. Conclusions: The findings indicate a slight increase in the rate of HIV testing among young people in the country but the rate of increase however too slow and fell far short of the standard required to achieve the UNAIDS first 90 and the national goal of testing 95% of people in the country by 2020.-this sentence is too long. Please break it up for easier reading and comprehension.

Response: revised.

3. A significant concern for this section is the lack of mention of age of consent and legal barriers to access to testing. Nigeria still has an applied age of consent to HIV testing of 18 years. There is however a 2014 National Consensus document that recommends age 14 yrs for the lower age of consent (Ref: Guidelines for Young Persons' Participation in Research and Access to Sexual and Reproductive Health Services in Nigeria). I strongly recommend a short narrative on this issue, making reference to the Child Rights Act, the 2014 consensus as above, some of the articles by Prof Morenike Folayan and colleagues on adolescent rights and access to SRH services, and the recent Bulletin of the WHO publication by McKinnon and Vandermorris: National age-of-consent laws and adolescent HIV testing in sub-Saharan Africa: a propensity-score matched study.

Response: we have added this paragraph.

Besides the individual level barriers, structural barrier also limits AYA’ access to HIV testing. For example, the legal age of consent for independent HIV testing is 18 years in Nigeria [41]. This law presents a structural barrier to accessing HIV testing services [42], as young people will mostly likely not bring their parents to provide consent and also providers would want to follow the law. The experts in the field of sexual and reproductive health (SRH) in Nigeria, during a strategic engagement held with the federal ministry of health, recommended age 14 yrs for the age of consent to HIV testing. We consider removing this legal barrier, as recommended by experts in the field of SRH in Nigeria, to be critical for expanding access to HIV testing in Nigeria.

---

## [Editor Report · Decision Letter 1]

5 May 2020

Low coverage of HIV testing among adolescents and young adults in Nigeria: implication for achieving the UNAIDS first 95

PONE-D-20-04813R1

Dear Dr. Ajayi,

We are pleased to inform you that your manuscript has been judged scientifically suitable for publication and will be formally accepted for publication once it complies with all outstanding technical requirements.

With kind regards,

Olanrewaju Oladimeji, MB;BS, Ph.D.

Academic Editor

PLOS ONE

Additional Editor Comments (optional):

Authors have addressed almost all the critical issues raised and the manuscript reads well
---

## [Editor Report · Acceptance letter]

8 May 2020

PONE-D-20-04813R1 

Low coverage of HIV testing among adolescents and young adults in Nigeria: implication for achieving the UNAIDS first 95 

Dear Dr. Ajayi:

I am pleased to inform you that your manuscript has been deemed suitable for publication in PLOS ONE. Congratulations! Your manuscript is now with our production department. 

With kind regards,

on behalf of

Dr. Olanrewaju Oladimeji 

Academic Editor

PLOS ONE